# Thermo-responsive gels that absorb moisture and ooze water

Kazuya Matsumoto[1], Nobuki Sakikawa[2] & Takashi Miyata [1,3]

The water content of thermo-responsive hydrogels can be drastically altered by small changes in temperature because their polymer chains change from hydrophilic to hydrophobic above their low critical solution temperature (LCST). In general, such smart hydrogels have been utilized in aqueous solutions or in their wet state, and no attempt has been made to determine the phase-transition behavior of the gels in their dried states. Here we demonstrate an application of the thermo-responsive behavior of an interpenetrating polymer network (IPN) gel comprising thermo-responsive poly(N-isopropylacrylamide) and hydrophilic sodium alginate networks in their dried states. The dried IPN gel absorbs considerable moisture from air at temperatures below its LCST and oozes the absorbed moisture as liquid water above its LCST. These phenomena provide energy exchange systems in which moisture from air can be condensed to liquid water using the controllable hydrophilic/hydrophobic properties of thermo-responsive gels with a small temperature change.

---

[1] Department of Chemistry and Materials Engineering, Kansai University, 3-3-35, Yamate-cho, Suita, Osaka 564-8680, Japan. [2] Health & Environment Systems Division, SHARP Corporation, 3-1-72, Kitakamei-cho, Yao, Osaka 581-8585, Japan. [3] Organization for Research and Development of Innovative Science and Technology, Kansai University, 3-3-35, Yamate-cho, Suita, Osaka 564-8680, Japan. Correspondence and requests for materials should be addressed to T.M. (email: tmiyata@kansai-u.ac.jp)

Hydrogels are attractive soft materials consisting of physically or chemically cross-linked polymer networks and aqueous solutions. A variety of hydrogels have been widely utilized as foods, disposal diaper, contact lenses, and so on because they exhibit fascinating behaviors such as water absorption, swelling, permeability, viscoelasticity, transparency, and biocompatibility. In addition, discovery of the volume phase transition of hydrogels led to the development of not only hydrogel science but also polymer science[1,2]. Some hydrogels have a unique property that they undergo abrupt changes in their volume in response to environmental changes, such as pH[3], temperature[4,5], electric field[6], light[7], and biomolecules[8,9]. Such stimuli-responsive hydrogels have many potential applications as smart and soft materials in aqueous solutions or in their wet state.

Thermo-responsive hydrogels such as poly(N-isopropylacrylamide) (PNIPAAm) hydrogels have attracted considerable attention in a variety of fields because they undergo a drastic change in volume in response to a slight change in temperature. PNIPAAm hydrogels, which have a lower critical solution temperature (LCST) of around 32 °C, drastically change their hydrophilicity/hydrophobicity at their LCST[4,5,10]. PNIPAAm hydrogels have been extensively studied for sensing[11], drug delivery[12–14], and cell cultures[15] because of their unique thermo-responsive properties. In general, the drastic changes in the hydrophilicity/hydrophobicity and volume of PNIPAAm hydrogels are observed in aqueous solutions. Although such thermo-responsiveness and various applications of PNPAAm hydrogels in aqueous media are well known, any researchers have never observed unique thermo-responsive behavior of the dried PNIPAAm gels in air. There have been few studies of the applications of the thermo-responsive behavior of PNIPAAm gels in their dried states.

Herein, we report that dried gels containing PNIPAAm can be applied as thermo-responsive moisture absorbents under relatively humid conditions. Surprisingly, the phase transition of water from the gas phase to the liquid phase was achieved by moisture absorption and water oozing through a drastic change in the hydrophilicity/hydrophobicity of the thermo-responsive gels by a small temperature change. This unique property demonstrates the possibility of applying thermo-responsive gels as high energy efficiency materials for condensing gaseous water to liquid water. It is common knowledge that the gas–liquid phase transition of water molecules occurs at 100 °C under a pressure of 1 atm (Figure 1a, left). The PNIPAAm hydrogel, which is a water-swollen state of the dried PNIPAAm gel, exhibits a volume phase transition at its LCST (Figure 1a, right). We found that the PNIPAAm gel undergoes a drastic change in its affinity for water in response to a slight change in temperature despite being in the dried state. It should be noted that the water molecules absorbed as gaseous water from the air oozed out as liquid water from the dried gel as a result of the drastic change in the hydrophilicity/hydrophobicity of the PNIPAAm chains (Figure 1a, center). Our concept includes a simple change of starting point from a wet state to a dried state in applications of thermo-responsive gels. By the simple change of the starting point, we can achieve the collection of moisture (gaseous water) as liquid water using thermo-responsive gels, followed by their innovative applications in their dried state.

## Results

### Preparation of thermo-responsive hydrogels and dried gels. As illustrated in Figure 1b, we designed an interpenetrating polymer network (IPN)[16,17] gel comprising PNIPAAm chains as a thermo-responsive component and sodium alginate (Alg) chains as a hydrophilic component to enhance the moisture absorption

capacity. After the first network was formed by copolymerization of N-isopropylacrylamide (NIPAAm) and N,N'-methylenebisacrylamide (MBAA), the IPN hydrogel of PNIPAAm and Alg was prepared by formation of a second network by ionic crosslinking of Alg with CaCl₂[18] within the first network. Copolymer hydrogels based on NIPAAm and sodium acrylate (AAcNa) were also prepared for comparison with the PNIPAAm/Alg IPN hydrogel. The LCST of the resulting copolymer (P(NIPAAm-co-AAcNa)) hydrogels in water increased with increasing AAcNa content and disappeared by introducing an AAcNa content of more than 5 mol% (Figure 2a). This means that the introduction of AAcNa component to enhance the moisture absorption capacity is limited to 5 mol%. In contrast, a clear LCST for the PNIPAAm/Alg IPN hydrogel was observed at 32 °C, which is the same as the LCST of PNIPAAm, despite it containing a hydrophilic Alg component. The introduction of the IPN structure enhanced the hydrophilicity of the networks without changing the LCST of PNIPAAm. As a result, the PNIPAAm/Alg IPN hydrogel exhibited a drastically thermo-responsive change in hydrophilicity/hydrophobicity with enhancing the moisture absorption capacity, differing from PNIPAAm-containing copolymer hydrogels. Therefore, by completely freeze-drying the PNIPAAm/Alg IPN hydrogel under vacuum after it being frozen at −20 °C, we prepared a thermo-responsive gel as a moisture absorbent under relatively humid conditions.

### Moisture absorption of dried IPN gels. After the PNIPAAm, Alg, and PNIPAAm/Alg IPN hydrogels were completely freeze-dried, moisture absorption experiments using the resultant gels were performed at constant temperature and relative humidity (RH) conditions (27 °C and 80% RH). The dried PNIPAAm, Alg, and PNIPAAm/Alg IPN gels absorbed 0.2 g, 1.2 g, and 0.6 g of moisture per 1 g, respectively (Figure 2b). The introduction of Alg improves the moisture absorption capacity of the dried gel. Figure 2c shows that the moisture absorption of the dried IPN gel decreases gradually with increasing temperature. This may be due to the change of the PNIPAAm chains from hydrophilic to hydrophobic. It should be noted that the moisture absorption isotherms above 35 °C were quite different from those below 30 °C. This demonstrates that the mechanism for moisture absorption by the IPN gel changes drastically at the LCST of PNIPAAm. From the moisture absorption isotherms below 30 °C, it can be observed that the absorption capacity of the IPN gel increased slowly at relative humidities of <60% RH but increased sharply above 60% RH. The moisture absorption isotherms below 30 °C and above 35 °C seem to be of the Brunauer–Emmett–Teller (BET) type[19] and the Langmuir type[20], respectively. In Figure 2d, the absorption isotherms at 25 and 35 °C are replotted using the BET equation. The moisture absorption isotherm at 25 °C can be satisfactorily fitted by linear regression based on the BET isotherm model. The weight of the IPN gel increased a little by moisture absorption and the dried gel became wet, meaning that the moisture absorption of the dried IPN gel can induce its slight swelling. A few researchers reported the swelling of hydrophilic polymers by the absorption of water vapors[21–23]. In the previous papers, furthermore, sorption isotherms of water vapor into hydrophilic polymers, which exhibited the swelling under high humidity condition, have been fitted to the BET model. Therefore, the moisture absorption isotherm at 25 °C shown in Fig. 2d indicates that multilayer absorption of water molecules onto the IPN gel occurs, which may be due to a slight swelling under high humidity conditions. Furthermore, the moisture absorption isotherm at 35 °C changed from BET type to Langmuir type, revealing different mechanisms for moisture absorption between 25 and 35 °C. Absorption of water vapor into the thermo-

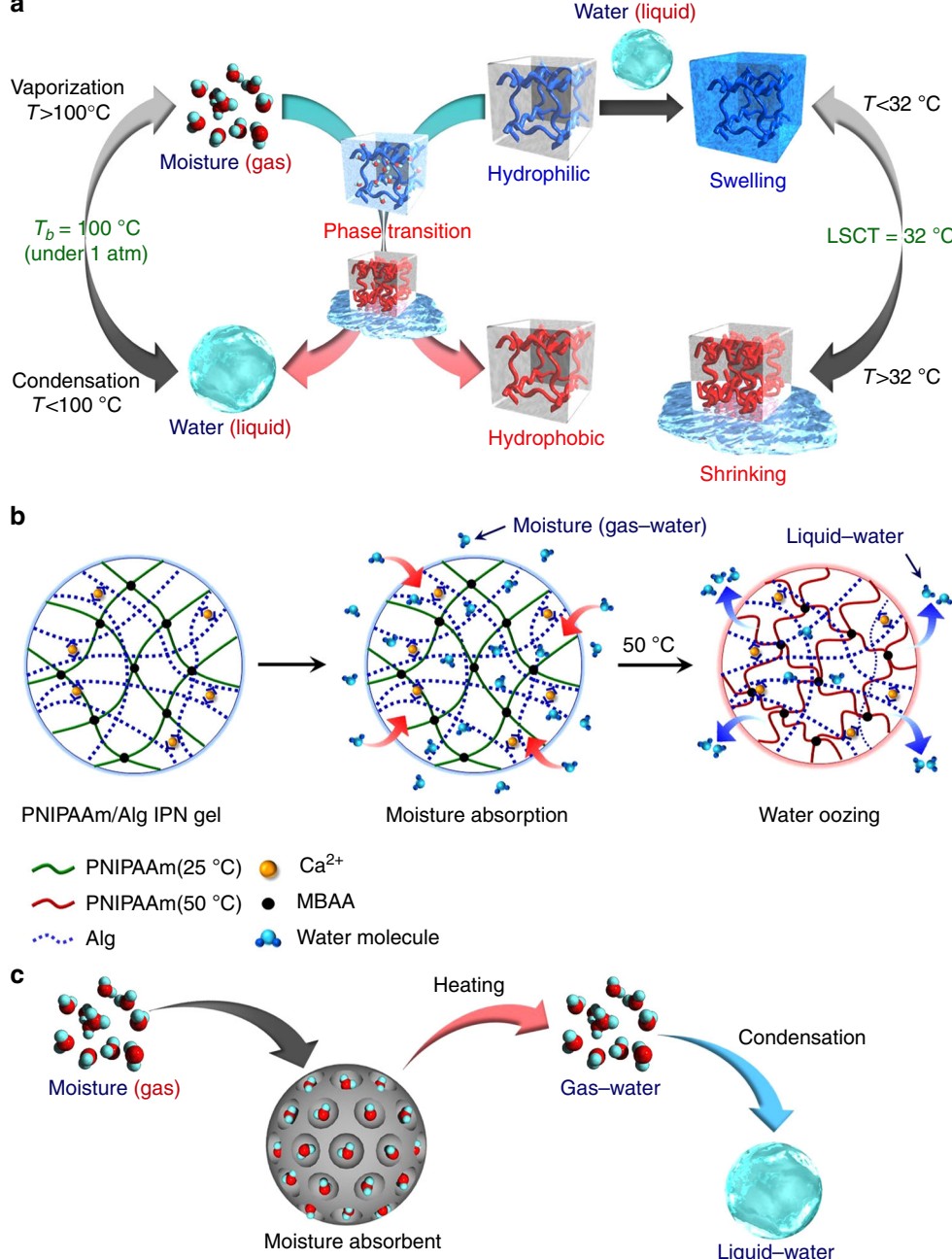

**Fig. 1** Conceptual illustration of this study. **a** Water-adsorption and oozing behavior of dried PNIPAAm/Alg IPN gel. **b** Moisture absorption and oozing behavior of IPN gels caused by temperature changes. **c** Condensation of moisture using a standard absorbent

responsive IPN gel above its LCST seems to be fitted to the Langmuir model because a small number of water molecules are adsorbed on its hydrophilic moieties, followed by the monolayer formation, similarly to the standard absorption into hydrophobic polymers. Below the LCST, however, a larger number of water molecules are absorbed into the IPN gel, allowing the slight swelling. The slight swelling results in an increase in the surface area or absorption capacity of the IPN gel. In addition, hydrophobic moieties of PNIPAAm networks may be covered with water molecules by their adsorption onto the polymer chains and the networks may become more hydrophilic. As a result, moisture absorption into the IPN gels below the LCST can be fitted to the BET model owing to the multilayer absorption. The difference in absorption mechanism results from a thermo-responsive change in the hydrophilicity/hydrophobicity of the PNIPAAm chains. As

a result, the moisture absorption of the IPN gel can be regulated by temperature but it is not a swollen hydrogel in water.

**Oozing behavior of water from IPN gel.** Using a digital microscope with low magnification, we monitored the surface of the PNIPAAm/Alg IPN gel during heating at 50 °C after moisture absorption at 25 °C. Liquid water appeared on the gel surface during heating at 50 °C, and the amount of water increased with increasing heating time (Figure 3 and Supplementary Movie 1). This demonstrates that the moisture (gaseous water) that was absorbed into the completely dried IPN gel oozed out as liquid water when the temperature was increased above the LCST of PNIPAAm. Because the IPN gel was completely dried before the moisture absorption, the fact that liquid water oozed out of the

gel indicates that gaseous water was converted to liquid water by the thermo-responsive gel. In general, water molecules need to be cooled to be converted from a gaseous state to a liquid state because their condensation is exothermic. The phase transition between a gaseous state and a liquid state is directly associated with the latent heat of vaporization or condensation. Therefore,

dehumidifiers involving a few steps, i.e., moisture absorption into absorbents, water collection, and regeneration of the absorbents, require high energy consumption to evaporate water for the regeneration and to condense it for water collection (Fig. 1c). Surprisingly, the thermo-responsive IPN gel enables water molecules to be condensed from a gaseous state to a liquid state

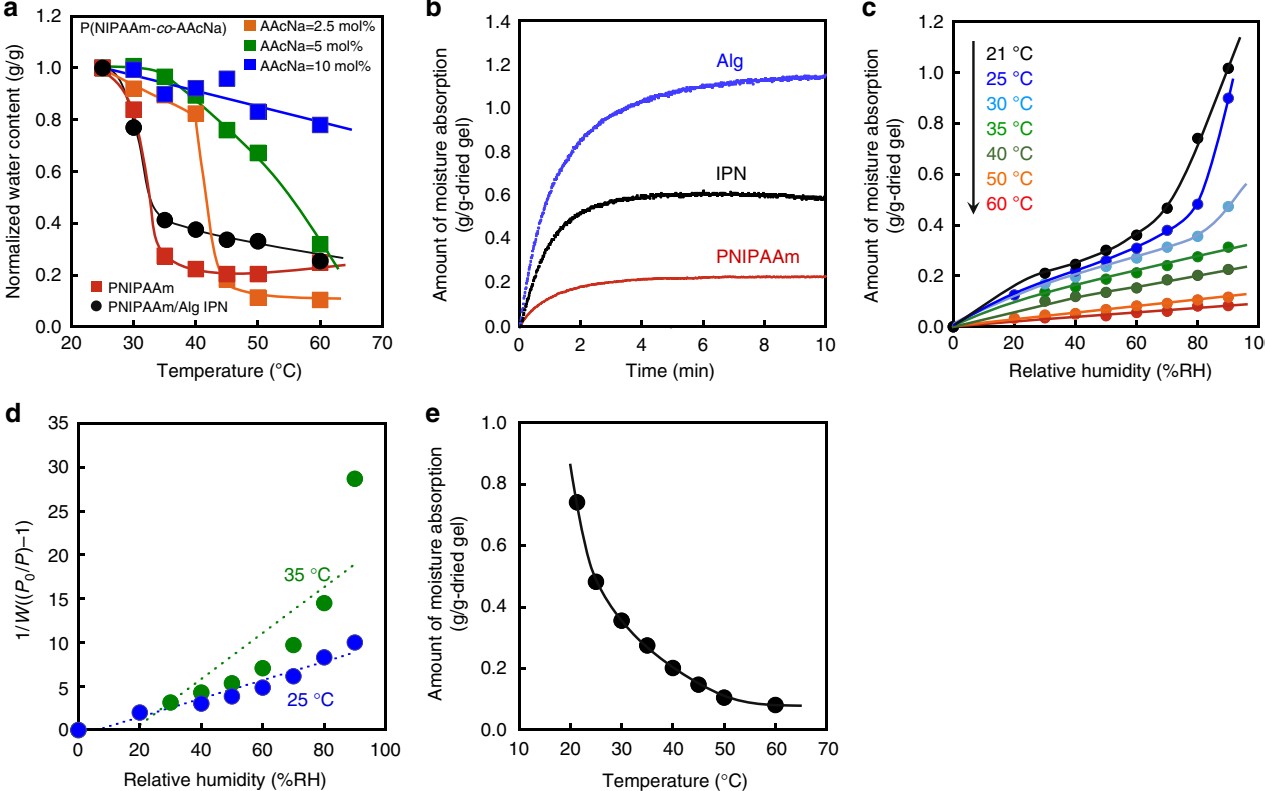

**Fig. 2** Thermo-responsive behaviors of hydrogels and dried gels. **a** Relationship between temperature and normalized water content of PNIPAAm hydrogel, P(NIPAAm-co-AAcNa) hydrogels with an AAcNa content of 2.5, 5, and 10 mol%, and PNIPAAm/Alg IPN hydrogel in water. **b** Moisture-absorption kinetics of the freeze-dried IPN, the Alg, and the PNIPAAm gels at 27 °C and 80% RH. **c** Moisture-absorption isotherms of the IPN gels at various temperatures. **d** BET plots of the IPN gels at 25 and 35 °C. **e** The absorption capacity of dried IPN gels at 80% RH and various temperatures

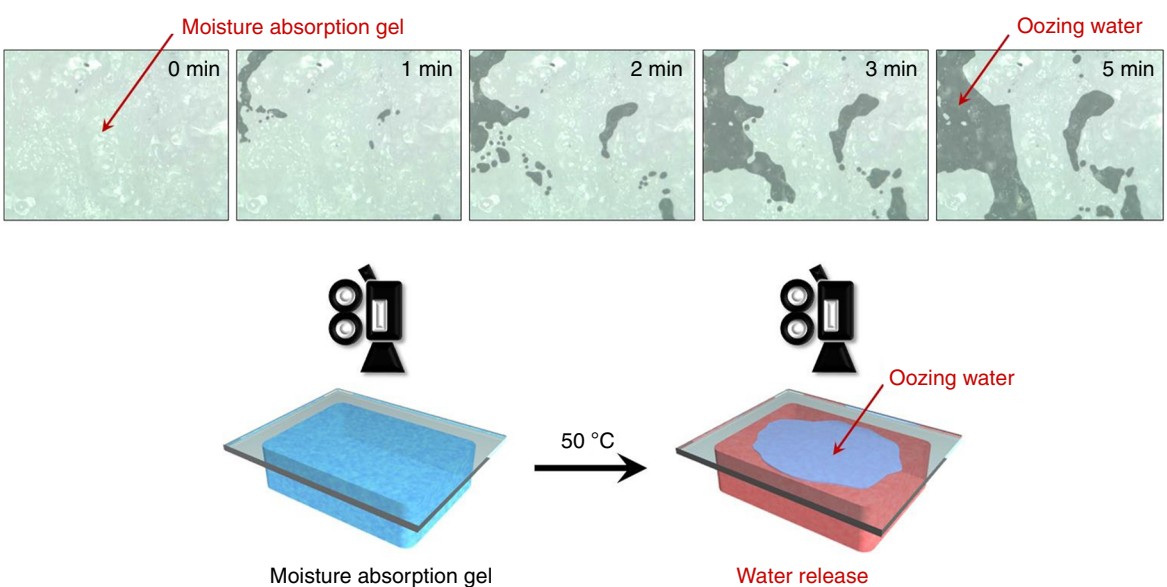

**Fig. 3** Images of thermo-responsive water oozing. Images of the water oozing behavior of the PNIPAAm/Alg IPN gel during heating at 50 °C after moisture absorption

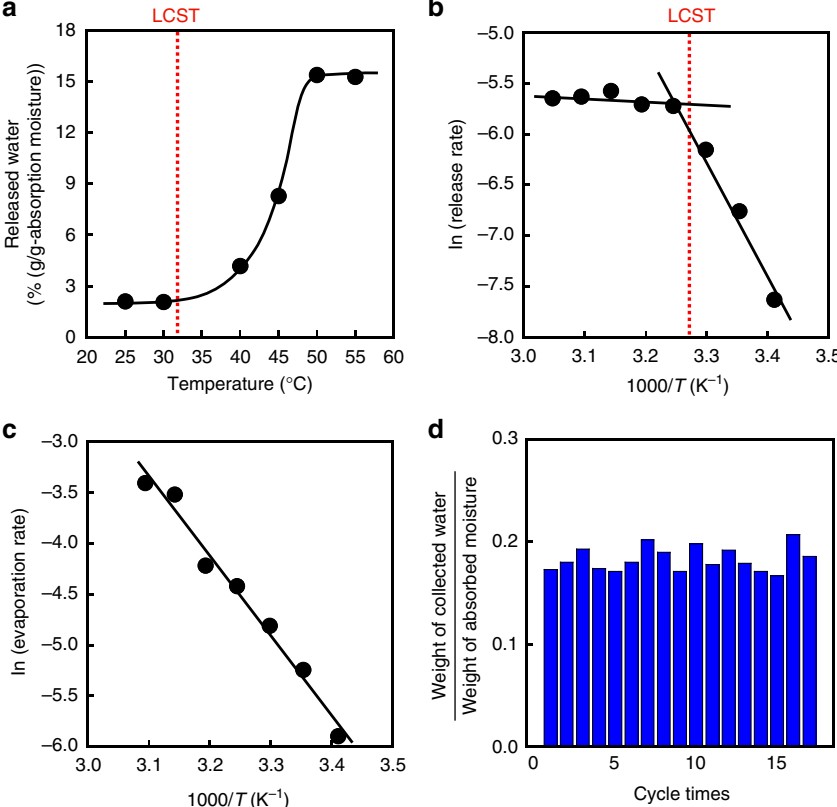

**Fig. 4** Thermo-responsive water oozing behaviors of IPN gels. **a** Relationship between temperature and the amount of water oozed from the PNIPAAm/Alg IPN gel. Plots of the natural logarithm of the rate versus the reciprocal of the absolute temperature for water oozing from (**b**) the dried IPN gels and (**c**) pure water. **d** Cycle tests of water oozing from the surface of the IPN gel at 50 °C after the moisture absorption at 27 °C and 80% RH

by a slight change in temperature from 25 to 50 °C. As a result, liquid water can be collected from moisture by an oozing process using the thermo-responsive gel with low energy consumption because this process requires no latent heat of evaporation and condensation for the absorbent regeneration and water collection. Taking into account that moisture absorption into the dried IPN gel changed sensitively in response to a slight temperature change, as shown in Figure 2e, the most likely explanation for water oozing from the gel is a drastic change in the hydrophilicity/hydrophobicity of the PNIPAAm chains above the LCST. To reveal a change in the hydrophilicity of the dried IPN gel, we measured water contact angles on the dried IPN gel at 25 and 50 °C. The IPN gel showed small water contact angles at 25 and 50 °C owing to hydrophilic Alg chains. The water contact angle on the IPN gel increased from 23.1 to 36.3° by rising temperature from 25 to 50 °C above the LCST (Supplementary Figure 1). The larger water contact angle at 50 °C than 25 °C demonstrates that the hydrophobicity of the dried IPN gel increased by rising temperature in air. Although hydrophilic Alg chains in the IPN gel prevent the water contact angle from increasing remarkably, a drastic change of PNIPAAm chains from hydrophilic to hydrophobic above the LCST causes an increase in water contact angle on the IPN gel in air. Therefore, water oozing from the IPN gel is mainly attributed to a drastic change in the hydrophilicity/hydrophobicity of the thermo-responsive PNIPAAm chains.

After moisture absorption into the dried PNPAAm/Alg IPN gel, we quantitatively investigated the oozing behavior as a function of temperature. The amount of liquid water that oozed out from the gel increased sharply with increasing temperature above 40 °C (Figure 4a). In addition, the initial release rate of

liquid water from the IPN gel was measured using thermogravimetry–differential thermal analysis (TG–DTA) equipped with a humidity generator. The Arrhenius plot for evaporation of pure water shown in Figure 4c yielded a straight line at temperatures ranging from 21 to 50 °C, revealing that evaporation without the gel is based on the temperature dependence of the vapor–liquid equilibrium. From the slope of the Arrhenius plot, the activation energy for the evaporation of water was determined to be 64.9 kJ mol$^{-1}$. In contrast, the release rate of water from the IPN gel changed drastically at temperatures close to the LCST of PNIPAAm (Figure 4b). The two straight lines on the Arrhenius plot for the IPN gel indicate that the mechanism for water release above the LCST was quite different from that below the LCST. The activation energy for water release below the LCST was 95.3 kJ mol$^{-1}$, which is similar to that for the evaporation of water. However, the activation energy for water release above the LCST decreased to as low as 3.8 kJ mol$^{-1}$, much lower than that below the LCST. Above the LCST, both oozing liquid water and evaporating vapor from the IPN gel contributed to the activation energy. The low activation energy above the LCST implies that the water release above the LCST was governed by oozing as a liquid water from the IPN gel although a part of water was evaporated from the gel similarly to the evaporation below the LCST. Oozing from the IPN gel above the LCST is attributed to a drastic change of the PNIPAAm from hydrophilic to hydrophobic. The low activation energy above the LCST demonstrates that we can extract liquid water from the gel after moisture absorption with low energy consumption. Thus, the thermo-responsive gel can induce a phase transition of water from the gas phase to the liquid phase in response to a slight temperature change.

**Repetitive moisture absorption and water oozing using IPN gel**. We performed cycle tests of moisture absorption and water oozing using the IPN gel because its applications to energy exchange systems such as dehumidifiers with low energy consumption require repetitive water oozing. The cycle tests demonstrated that moisture absorbed into the IPN gel repeatedly oozed out as liquid water during cyclic changes between the temperature below and above the LCST (Figure 4d). In the cycle tests using the IPN gel, we collected ~20% water as a liquid state to the totally absorbed moisture. At a temperature above the LCST, because PNIPAAm chains change from hydrophilic to hydrophobic but Alg chains are hydrophilic, hydrophilic Alg chains prevented water from being released out of the IPN gel. This is the reason why absorbed water partially remains in the IPN gel after water oozing out above the LCST. As a result, the repetitive moisture absorption and water oozing provide an energy exchange system with low energy consumption in which moisture from air can be condensed to liquid water using the repeatedly controllable hydrophilic/hydrophobic properties of thermo-responsive gels by a small temperature change.

## Discussion

In this study, we prepared the PNIPAAm/Alg IPN gel for the collection of moisture (gaseous water) as liquid water by a small temperature change, followed by innovative applications of thermo-responsive gels in their dried state. Repetitively the dried IPN gel absorbed considerable moisture from air at temperatures below its LCST and oozed the absorbed moisture as liquid water above its LCST. From the discussions about moisture absorption using Langmuir and BET type models and about the activation energy for water release above the LCST, we can explain the mechanism for the absorption of moisture into the dried IPN gel and oozing it as liquid water as follows. During the moisture absorption into the IPN gel below the LCST, water molecules are adsorbed on both Alg and PNPAAm chains similarly to the moisture absorption into general hydrophilic polymers[21–23], followed by a slight swelling of the IPN gel. The slight swelling results in hydrophobic hydration of PNIPAAm chains although the gel network is not immersed in an aqueous solution. It is well known that PNIPAAm undergoes a drastic change from hydrophilic to hydrophobic in aqueous solutions with rising temperature above the LCST because the negative entropy of water molecules around the nonpolar regions of PNIPAAm chains dominates. Similarly, the PNIPAAm chains in the IPN gel slightly swollen by moisture absorption change from hydrophilic to hydrophobic by the dehydration based on the negative entropy of water molecules around the PNIPAAm chains with rising temperature. As the PNIPAAm chains become hydrophobic above the LCST, the water molecules on the PNIPAAm chains are desorbed to be condensed as liquid water. Thus, the moisture absorbed into the thermo-responsive IPN gel oozes out as liquid water above the LCST. In contrast, the water molecules adsorbed on the Alg chains are preserved within the IPN gel owing to their high hydrophilicity despite a temperature above the LCST of PNIPAAm. Desorption of water molecules from the PNIPAAm chains results in a slight shrinkage of the gel network. Water molecules on the Alg chains may be squeezed out by the slight shrinkage of the network with oozing water from the PNIPAAm chains. As a result, the thermo-responsive IPN gel can absorb moisture from air and oozing it as liquid water by cycle changes in temperature.

In summary, both moisture absorption and water oozing were achieved using the thermo-responsive IPN gel. The thermo-responsive change in the hydrophilicity/hydrophobicity of the IPN gel enables water molecules to be converted from a gaseous state to a liquid state by a slight change in temperature above the LCST. The moisture absorption and water oozing behaviors of the thermo-responsive gel can provide a useful tool for constructing energy exchange systems with high efficiency such as dehumidifiers with low energy consumption[24–27]. Although the oozing of liquid water from the IPN gel still requires further research for improving the efficiency, moisture absorption and water oozing using thermo-responsive gels is likely to become quite an innovative system in the future. The concept of the phase transition of water molecules from the gas phase to the liquid phase using a thermo-responsive gel in its dried state will open up several possibilities for developing soft materials for energy exchange systems with low energy consumption.

## Methods

**Preparation of IPN hydrogel and freeze-dried gels**. PNIPAAm/Alg IPN gel was prepared as follows: NIPAAm (700 mg, 6.19 mmol) as a main monomer and MBAA (85 mg, 0.55 mmol) as a chemical cross-linker were dissolved in 23 mL of water. After Alg (700 mg, 1.4 mmol) was added to the solution, it was stirred at room temperature overnight. After the addition of 1 mL of 0.1 M aqueous ammonium peroxodisulfate (APS) solution and 1 mL of 0.8 M aqueous $N,N'$-tetramethylethylenediamine (TEMED) solution as redox initiators, polymerization was performed in an ice bath for 6 h. The resultant hydrogel was immersed in 0.25 M aqueous calcium chloride solution for ionic cross-linking of the Alg chains with calcium ions. A standard PNIPAAm hydrogel was also prepared by copolymerization of NIPAAm (700 mg) and MBAA (85 mg) in 23 mL of water using 1 mL of 0.1 M aqueous APS solution and 1 mL of 0.8 M aqueous TEMED solution. A standard Alg hydrogel was prepared by adding 0.25 M aqueous calcium chloride solution to 2.7 wt% aqueous Alg solution. The resultant PNIPAAm/Alg IPN, PNIPAAm, and Alg hydrogels were frozen at −20 °C and dried completely under vacuum.

**Preparation of P(NIPAAm-co-AAcNa) hydrogels**. NIPAAm, AAcNa, and MBAA were dissolved in 23 mL of water. After 1 mL of 0.1 M aqueous APS solution and 1 mL of 0.8 M aqueous TEMED solution were added to the solution, copolymerization was performed in an ice bath for 6 h. After the copolymer (P(NIPAAm-co-AAcNa)) hydrogels were formed with various mole percents of AAcNa in the feed, they were immersed in water in order to remove unreacted monomers.

**Swelling measurements in water**. The normalized water contents of the PNIPAAm/Alg IPN and P(NIPAAm-co-AAcNa) hydrogels in aqueous media were measured gravimetrically under certain conditions after the excess aqueous calcium chloride solution or water on the hydrogel surface was gently removed with filter paper. The hydrogels were incubated for at least 12 h at 25, 30, 35, 40, 45, 50, and 60 °C. The normalized water content of the hydrogel is defined as follows:

$$\text{Normalized water content} = W/W_0 \qquad (1)$$

where $W$ is the weight of water in the swollen hydrogel at each temperature and $W_0$ is the weight of the swollen hydrogel at 25 °C.

**Evaluation of moisture absorption**. The absorption capacity of the dried PNIPAAm/Alg IPN gel was investigated by mass measurements under constant, predetermined conditions of temperature and RH. The samples were heated in flowing $N_2$ (TG-DTA; 250 mL min$^{-1}$) at different partial pressures of water vapor, which were controlled by a humidity generator TG-DTA/HUM-1 (Rigaku Corp.). The absorption capacity was calculated from the increase in mass of the dried gels after equilibration at a given RH.

The BET model was used to analyze the moisture absorption data. The BET equation can be expressed as follows:

$$\frac{1}{v\left[\left(\frac{P_0}{P}\right)-1\right]} = \frac{C-1}{v_m C}\left(\frac{P}{P_0}\right) + \frac{1}{v_m C} \qquad (2)$$

where $P/P_0$ is the RH, $v$ is the amount of absorbed moisture, $v_m$ is the saturation moisture content of a monolayer, and $C$ is the BET constant, which relates to the absorption energy.

**Evaluation of oozing behavior**. Oozing behavior from the PNIPAAm/Alg IPN gel after moisture absorption was observed using a digital microscope (VW-9000, KEYENCE Co. Ltd., Japan). After the PNIPAAm/Alg IPN gel was covered with a glass plate, it was heated to 50 °C using a Peltier device. The amount of water that oozed from the IPN gels was determined by measuring the weight of a filter paper that was sandwiched between the IPN gel and the glass plate. An analysis of the

oozing rate was performed with the assumption of simple first-order kinetics and an Arrhenius equation defined as follows:

$$k = A \exp\left(-\frac{E_a}{RT}\right) \tag{3}$$

where $k$ is the rate constant, $A$ is the pre-exponential factor, $E_a$ is the activation energy, $R$ is the universal gas constant, and $T$ is the absolute temperature. The activation energy was determined from the slope of a straight line fit to the Arrhenius plot.

**Cycle tests of moisture absorption and water oozing**. Cycle tests for moisture absorption and water oozing were performed using the dried PNIPAAm/Alg IPN gel with a diameter of 70 mm and a thickness of 1.5 mm as follows. First of all, moisture was absorbed into the dried PNIPAAm/Alg IPN gel at 27 °C and 80% RH. After the moisture absorption for 60 min, the IPN gel was placed on a plate heater at 50 °C. Then the water oozed from the surface of the IPN gel was transferred to the surface of a glass plate and its amount was determined by measuring a change in the weight of the glass plate, completing one full test cycle. This procedure was repeated 17 times with the same IPN gel.

**Data availability**. The authors declare that all data supporting the findings of this study are available within the article and its Supplementary Information or from the corresponding author upon reasonable request.

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

## Acknowledgements

We thank Mr. Yosuke Natsume (Kansai University) for his assistance in the contact angle measurements.

## Author contributions

N.S. and T.M. conceived and designed the experiments. T.M. supervised the project. K. M. and N.S. performed the experiments. K.M., N.S, and T.M. analyzed the experimental results. K.M. and T.M. wrote the manuscript. T.M. edited and revised the manuscript.

## Additional information

**Competing interests:** N.S. is a current employee of Sharp Corporation, and the remaining authors declare no competing interests.

