## [Peer Review File · Nature Communications]

Reviewers' comments:

Reviewer #1 (Remarks to the Author):

In this work the authors describe the ability of hydrogels based on interpenetrated networks of PNIPAAm to absorb moisture and ooze above the Low critical solution temperature of the network. The manuscript is carefully written and the results are shown in a well organize fashion. However, the novelty of the scientific case here is not demonstrated. The ability of water resealing in hydrogels is in fact, the main basis for its applicability as drug deliverers, and it is known for long time (see for example Journal of Controlled Released 2, 277- 288 (1985).

It is my impression that this is a careful and well performed study of the kinetics of water uptake and release of an hydrogel, but it is not suitable for being published in Nature Communications.

Reviewer #2 (Remarks to the Author):

This manuscript reports the study on a novel energy exchange system composed of thermo-responsive IPN gel which absorbs moisture and oozes water with low energy consumption. The main finding of this manuscript is that the dried IPN gel comprising thermo-responsive material and hydrophilic material absorbs moisture from air at temperatures below its LCST and oozes the absorbed moisture as liquid water above its LCST. The findings could be of interest to the energy generation, exchange, and storage communities. However, although an interesting thermo-responsive IPN gel is introduced, the detailed working mechanism of water phase change with dried IPN gel near LCST is not verified. There are many issues to be discussed to corroborate the authors' claims. Thus, the reviewer recommends the publication of this manuscript in Nature Communications after major revisions. The detailed comments are as follows.

1. PNIPAAm/Alg IPN hydrogels undergo volume change and hydrophilicity in response to a slight change of temperature, which absorbs moisture and oozes water. It is needed to explain the water condensation mechanism inside thermo-responsive hydrogels in molecular scale. Furthermore, why did hydrogels release water molecules as a liquid state, not as a vapor state?

2. Water condensation is exothermic, not endothermic as described in the manuscript. Furthermore, the activation energy calculated in the manuscript is not reasonable for the evaporation because the obtained activation energy is for the liquefied, released water. The authors need to think the phenomena carefully and discuss again with a different way.

3. The authors claimed that PNIPAAm/Alg IPN hydrogel follows BET model under LCST due to a slight swelling under high humidity conditions. How can a slight swelling of hydrogel induce multilayer adsorption of water molecules? How much does the dried hydrogel swell at each humidity condition as compared to that of hydrogel in water?

4. How can the thermo-responsive surface energy variation generate the transition between two different adsorption mechanisms? BET theory assumes that gas molecules only interact with adjacent layer. That is, the surface energy variation in the gel may contribute little to the development of multilayer adsorption. The authors need to explain about it.

5. How much is the oozed water as compared to total absorbed water? Is that amount enough for dehumidifying applications? Present associated data and ensure the stability of the hydrogels by repeated absorbing/oozing test.

6. What are the dimensions of the tested hydrogels? Hydrophobic/hydrophilic property is near-surface properties. Is it possible to scale-up the hydrogels in bulk without energy exchange efficiency?

7. In Figure 2a, it is difficult to understand comparison targets. The authors need to unify the expressions and clarify the comparison targets in the manuscript and Figure 2a.

Reviewer #3 (Remarks to the Author):

In this work, the authors developed novel water gas condensation system by interpenetrating polymer network consisting of thermo-responsive polymer. Although the NIPAAm-based hydrogels were well investigated, this system has never been developed. This system is a unique and new type of gas condensation system, and has potential impact to broad range of readers. However, one important point remains to be untackled in the current version; that is, repetitive absorption-oozing cycle. When the authors confirm the ability of repetitive absorption-oozing cycle, I recommend it for publication in Nature Communications.

Another minor point is P(NIPAAm-co-AAc) gels in Figure 2a. Because there are few description on these samples, I recommend remove them or add more experiments and discussion on these samples.

<Corresponding explanation for the reviewer's comments (Reviewer #1)>

Thank you very much for your kind comments on our manuscript. We revised it under your comments. We would like to explain which changes according to the reviewer's comments were made as follows:

Reviewer's comment (1):

In this work the authors describe the ability of hydrogels based on interpenetrated networks of PNIPAAm to absorb moisture and ooze above the Low critical solution temperature of the network. The manuscript is carefully written and the results are shown in a well organize fashion. However, the novelty of the scientific case here is not demonstrated. The ability of water resealing in hydrogels is in fact, the main basis for its applicability as drug deliverers, and it is known for long time (see for example Journal of Controlled Released 2, 277- 288 (1985).

It is my impression that this is a careful and well performed study of the kinetics of water uptake and release of an hydrogel, but it is not suitable for being published in Nature Communications.

Response:

As commented by the reviewer #1, the ability of water resealing in thermo-responsive hydrogels and their applicability as drug deliverers have been reported. It is well-known that thermo-responsive hydrogels undergo changes in the volume and water content in response to a small change in temperature. However, the changes in the volume and water content of the thermo-responsive hydrogels are usually evaluated in aqueous media. No researcher has observed unique thermo-responsive behavior of the dried gels that can absorb moisture and ooze water in air. Nobody has succeeded in utilizing thermo-responsive behavior of the gels in a dried state. This is the first paper describing innovative applications of thermo-responsive gels in the dried states. We thoroughly quantified this phenomenon in our model system by measuring the temperature dependence of the swelling ratio, water adsorption isotherms, absorption kinetics, and amount of water oozed from the gels. Although in the early stages of research, we believe that the temperature-controlled water absorption/oozing concept introduced here could eventually be used to develop innovative high-efficiency energy exchange systems, such as dehumidifiers. To clarify the originality of this study and its different point from many previous studies about thermo-responsive hydrogels in aqueous media, we added the following sentence (p. 3, lines 10-13):

"Although such thermo-responsiveness and various applications of PNIPAAm hydrogels in aqueous media are well-known, any researchers have never observed unique thermo-responsive behavior of the dried PNIPAAm gels in air."

Thank you again for your comments on our paper. We considered your kind advices and revised our manuscript. I hope this revision will meet your requirements satisfactory.

<Corresponding explanation for the reviewer's comments (Reviewer #2)>

Thank you very much for your kind comments on our manuscript. We revised it under your comments. We would like to explain which changes according to the reviewer's comments were made as follows:

Reviewer's comment (1):

PNIPAAm/Alg IPN hydrogels undergo volume change and hydrophilicity in response to a slight change of temperature, which absorbs moisture and oozes water. It is needed to explain the water condensation mechanism inside thermo-responsive hydrogels in molecular scale. Furthermore, why did hydrogels release water molecules as a liquid state, not as a vapor state?

Response:

Absorption of moisture into the thermo-responsive gels can be explained by the standard mechanism for absorption of water vapor into hydrophilic polymers. As described in our response for the reviewer's comment (3), a few researchers have studied the absorption behaviors of water vapor into hydrophilic polymers (Passauer, L., Struch, M., Schuldt, S., Appelt, J., Schneider, Y., Jaros, D. & Rohm, H. ACS Appl. Mater. Interfaces 4, 5852–5862 (2012); Hakalahti, M., Faustini, M., Boissière, C., Kontturi, E. & Tammelin T. Biomacromolecules, 18, 2951–2958 (2017)). According to the suggestion by the reviewer, we added the references (Ref. 16-18) and the description about the water condensation inside thermo-responsive hydrogels in molecular scale, based on the reference papers, in the text (p. 4, lines 18-24; lines 28-36):

"The weight of the IPN gel increased a little by moisture absorption and that the dried gel became wet, meaning that the moisture absorption of the dried IPN gel can induce its slight swelling. A few researchers reported the swelling of hydrophilic polymers by the absorption of water vapors¹⁶⁻¹⁸. In the previous papers, furthermore, sorption isotherms of water vapor into hydrophilic polymers, which exhibited the swelling under high humidity condition, have been fitted to the BET model. Therefore, the moisture absorption isotherm at 25 °C shown in Figure 2d indicates that multilayer absorption of water molecules onto the IPN gel occurs, which may be due to a slight swelling under high humidity conditions." (p. 4, lines 18-24)

"Absorption of water vapor into the thermo-responsive IPN gel above its LCST seems to be fitted to the Langmuir model because a small amount of water molecules are adsorbed on the hydrophilic moieties, followed by the monolayer formation, similarly to the standard absorption into hydrophobic polymers. Below the LCST, however, a larger amount of water molecules is absorbed into the IPN gel, allowing the slight swelling. The slight swelling results in an increase in the surface area or absorption capacity of the IPN gel. In addition, hydrophobic moieties of PNIPAAm networks may be covered with water molecules by their adsorption onto the polymer chains and the networks may become more hydrophilic. As a result, moisture absorption into the IPN gels below LCST can be fitted to the BET model owing to the multilayer absorption." (p. 4, lines 28-36)

In addition, the reviewer's question about the reason why the gels release water molecules as a liquid state, not as vapor state is very important. Our description may have misled the reviewer. Of course, water absorbed into the IPN gels was partially released as a vapor state, based on vapor–liquid equilibrium under the condition. Using the thermo-responsive IPN gels, most water molecules were directly collected as a liquid state although water was partially evaporated by vapor–liquid equilibrium. Oozing water as a liquid

state is very useful in the developments of innovative high-efficiency energy exchange systems such as dehumidifiers because the energy for water condensation from a vapor state is not required. In the discussion, we revised the sentences to avoid the misleading as follows (p. 5, lines 35-39):

"Above the LCST, both oozing liquid water and evaporating vapor from the IPN gel contributed to the activation energy. The low activation energy above the LCST implies that the water release above the LCST was governed by oozing as a liquid water from the IPN gel although a part of water was evaporated from the gel similarly to the evaporation below LCST. Oozing from the IPN gel above the LCST is attributed to a drastic change of the PNIPAAm from hydrophilic to hydrophobic."

Reviewer's comment (2):

Water condensation is exothermic, not endothermic as described in the manuscript. Furthermore, the activation energy calculated in the manuscript is not reasonable for the evaporation because the obtained activation energy is for the liquefied, released water. The authors need to think the phenomena carefully and discuss again with a different way.

Response:

As pointed out by the reviewer, water condensation is exothermic. The description in the manuscript is our mistake. We corrected it by exchanging the word "endothermic" with the word "exothermic"(p. 5, line 9). In addition, as commented by the reviewer, the activation energy described in the manuscript is for the liquefied and released water. As described in our response for the reviewer's comment (1), we revised the discussion based on the activation energy (p. 5, lines 35-39):

Reviewer's comment (3):

The authors claimed that PNIPAAm/Alg IPN hydrogel follows BET model under LCST due to a slight swelling under high humidity conditions. How can a slight swelling of hydrogel induce multilayer adsorption of water molecules? How much does the dried hydrogel swell at each humidity condition as compared to that of hydrogel in water?

Response:

The phrase "a slight swelling" of the dried IPN gel might confuse the reviewer. As shown in Figure 2c, below LCST, the amount of moisture absorption into the IPN gel increased sharply under high humidity conditions. Then the weight of the gel increased a little by moisture absorption and the dried gel became wet. Although the water content of the wet IPN gel is much lower than that of the hydrogel swollen in aqueous media, the moisture absorption into the dried IPN gel can induce an increase in the weight and its slight swelling. In addition, a few researchers reported the swelling of hydrophilic polymers by the absorption of water vapors (Passauer, L., Struch, M., Schuldt, S., Appelt, J., Schneider, Y., Jaros, D. & Rohm, H. *ACS Appl. Mater. Interfaces* **4**, 5852–5862 (2012); C. Galvin, J. & Genzer, J. *Macromolecules* **49**, 4316-4329 (2016); Hakalahti, M., Faustini, M., Boissière, C., Kontturi, E. & Tammelin T. *Biomacromolecules*, **18**, 2951–2958 (2017)). For example, by spectroscopic ellipsometry measurements under controlled humidity conditions, Passauer et. al. demonstrated the swelling of hydrophilic polymer brushes by water vapor in air. In the previous papers, furthermore, sorption isotherms of water vapor into hydrophilic polymers, which exhibited the swelling under high humidity condition, have been fitted to the BET model. Thus, moisture absorption into the dried polymers induces their slight swelling generally.

As shown in Figure 2c, the maximum weight of moisture absorbed into the dried IPN gel was approximately 1.0 g per 1 g of dried gel. On the other hand, the water content of the hydrogel swollen in water was approximately 94 wt%, meaning that the weight of water absorbed in the hydrogel was approximately 16 g per 1 g of dried gel. These means that the swelling ratio of the dried gel was much lower than that of hydrogel in water. The moisture absorption into the dried gel is based on the vapor–liquid equilibrium of water, whereas the swelling of the hydrogel in water results from the diffusion of polymer networks and water molecules in a large amount of water. Therefore, we think that the swelling of the dried gel by moisture absorption is quite different from that of hydrogel in water. As a result, the phrase "a slight swelling" may confuse the readers. Therefore, we added the references (Ref. 16-18) and revised the discussion about BET model with referring the previous paper as follows (p. 4, lines 18-26).

"The weight of the IPN gel increased a little by moisture absorption and that the dried gel became wet, meaning that the moisture absorption of the dried IPN gel can induce its slight swelling. A few researchers reported the swelling of hydrophilic polymers by the absorption of water vapors¹⁶⁻¹⁸. In the previous paper, furthermore, sorption isotherms of water vapor into hydrophilic polymers, which exhibited the swelling under high humidity condition, have been fitted to the BET model. Therefore, the moisture absorption isotherm at 25 °C shown in Figure 2d indicates that multilayer absorption of water molecules onto the IPN gel occurs, which may be due to a slight swelling under high humidity conditions."

Reviewer's comment (4):

How can the thermo-responsive surface energy variation generate the transition between two different adsorption mechanisms? BET theory assumes that gas molecules only interact with adjacent layer. That is, the surface energy variation in the gel may contribute little to the development of multilayer adsorption. The authors need to explain about it.

Response:

As described in our response for the reviewer's comment (3), in general, the sorption isotherms of water vapor into hydrophilic polymers, which exhibited the swelling under high humidity condition, have been fitted to the BET model. Absorption of water vapor into the thermo-responsive IPN gel above its LCST seems to be fitted to the Langmuir model because a small number of water molecules are adsorbed on the hydrophilic moieties, followed by the monolayer formation, similarly to the standard adsorption into hydrophobic polymers. Below the LCST, however, a larger number of water molecules are absorbed into the IPN gel, allowing the slight swelling. The slight swelling results in an increase in the surface area or absorption capacity of the IPN gel. In addition, hydrophobic moieties of PNIPAAm networks may be covered with water molecules during their adsorption onto the polymer chains and the networks may become more hydrophilic. As a result, moisture absorption into the IPN gels below LCST can be fitted to the BET model owing to the multilayer absorption. We understand that the moisture absorption and water oozing of the thermo-responsive IPN gel may still require further research to clarify the detailed mechanism responsible for its unique behavior in molecular scale, and this paper is a first step for the applications of thermo-responsive gels in a dried state. Therefore, with referring the reference papers (Ref. 16-18), we explain a drastic change in adsorption mechanisms from Langmuir type to BET type by rising temperature as describe in our response for the reviewer's comment (3) (p. 4, lines 18-26) and added the following sentences (p. 4, lines 28-36):

"Absorption of water vapor into the thermo-responsive IPN gel above its LCST seems to be fitted to the Langmuir model because a small number of water molecules are adsorbed on its hydrophilic moieties, followed by the monolayer formation, similarly to the standard absorption into hydrophobic polymers. Below the LCST, however, a larger number of water molecules are absorbed into the IPN gel, allowing the slight swelling. The slight swelling results in an increase in the surface area or absorption capacity of the IPN gel. In addition, hydrophobic moieties of PNIPAAm networks may be covered with water molecules by their adsorption onto the polymer chains and the networks may become more hydrophilic. As a result, moisture absorption into the IPN gels below LCST can be fitted to the BET model owing to the multilayer absorption."

Reviewer's comment (5):

How much is the oozed water as compared to total absorbed water? Is that amount enough for dehumidifying applications? Present associated data and ensure the stability of the hydrogels by repeated absorbing/oozing test.

Response:

As commented by the reviewer, the amount of water oozed from the gel should be compared to total amount of water absorbed into the gel. We collected water from oozed as a liquid state from the IPN gel by transferring to the surface of a glass plate, as described in "Cycle tests of moisture absorption and water oozing" of Methods. Completely collecting oozed water by the method was difficult, but approximately 10% water to the totally absorbed water was collected as a liquid state from the gel. Even if temperature is above LCST, water molecules can be adsorbed on the hydrophilic moieties of the gel networks. In addition, water was partially released as a vapor state from the gel, based on vapor–liquid equilibrium. As a result, it is difficult for the gel to release all water molecules as a liquid state and for us to collect them completely. We don't think that the amount of water oozed from the gel is still enough for dehumidifying applications now. Therefore, we have started the project for the improvement of the efficiency of absorbing/oozing water using thermo-responsive gels. This paper is the first step to demonstrate the applications of thermo-responsive gels in a dried state. Even though the moisture absorption and water oozing using the thermo-responsive gels still require further research work into possible applications as dehumidifiers, they are likely to become quite innovative systems in the future. As pointed out by the reviewer, repetitive absorbing/oozing test is very important in the development of innovative high-efficiency energy exchange systems. We performed the repetitive cycle tests and added the results and discussion in the main text (p. 6, lines 3-7) and in Methods (p. 8, lines 24-30), together with Figure 4 (d):

In the main text (p. 6, lines 3-7):

"In addition, we performed cycle tests of moisture absorption and water oozing using the IPN gel because its applications to energy exchange systems such as dehumidifiers with low energy consumption require repetitive water oozing. The cycle tests demonstrated that moisture absorbed into the IPN gel repeatedly oozed out as liquid water during cyclic changes between the temperature below and above LCST (Figure 4d)."

In Methods (p. 8, lines 24-31):

"Cycle tests of moisture absorption and water oozing

Cycle tests for moisture absorption and water oozing were performed using the dried PNIPAAm/Alg IPN gel with a diameter of 70 mm and a thickness of 1.5 mm as follows. First of all, moisture was absorbed

into the dried PNIPAAm/Alg IPN gel at 27 °C and 80% RH. After the moisture absorption for 60 min., the IPN gel was placed on a plate heater at 50 °C. Then the water oozed from the surface of the IPN gel was transferred to the surface of a glass plate and its amount was determined by measuring a change in the weight of the glass plate, completing one full test cycle. This procedure was repeated 15 times with the same IPN gel."

Reviewer's comment (6):

What are the dimensions of the tested hydrogels? Hydrophobic/hydrophilic property is near-surface properties. Is it possible to scale-up the hydrogels in bulk without energy exchange efficiency?

Response:

As pointed out by the reviewer, the dimension of the thermo-responsive gels is an important factor to govern their energy exchange efficiency in developments of innovative dehumidifiers. In addition, specific surface area is also very important because the surface area of gels determines the rate of moisture absorption and water oozing. We have started a new project to design thermo-responsive gels with a large specific surface area for improving the rate of moisture absorption and water oozing. Now the effect of the dimension and surface area of the thermo-responsive gels on their moisture absorption and water oozing is under the investigations. In this manuscript, we focus on only unique behaviors of the moisture absorption and water oozing of thermo-responsive gels in a dried state because no paper has been reported on potential applications of the thermo-responsive gels in a dried state. In the revised manuscript, we revised the main text (p. 6, lines 1-5) and added the sentence about the dimension of the thermo-responsive gels in Methods (p. 8, lines 25-26) as follows:

In the main text (p. 6, lines 13-16):

"Although the oozing of liquid water from the IPN gel still requires further research for improving the efficiency, moisture absorption and water oozing using thermo-responsive gels is likely to become quite an innovative system in the future."

In Methods (p. 8, lines 25-26)

"Cycle tests for moisture absorption and water oozing were performed using the dried PNIPAAm/Alg IPN gel with a diameter of 70 mm and a thickness of 1.5 mm as follows."

Reviewer's comment (7):

In Figure 2a, it is difficult to understand comparison targets. The authors need to unify the expressions and clarify the comparison targets in the manuscript and Figure 2a.

Response:

We understood that it might be difficult to understand comparison targets in Figure 2a. In Figure 2, we compare a IPN hydrogel, which were composed of thermo-responsive PNIPAAm and hydrophilic Alg, with copolymer hydrogels from NIPAAm and acrylic acid component (AAcNa). Figure 2a demonstrates that the formation of IPN structure enables us to introduce a large amount of hydrophilic components into the thermo-responsive gel without reducing thermo-responsiveness. According to the reviewer's comment, we revised Figure 2a and the description to unify the expressions and to clarify the comparison target in the revised manuscript (p. 3, line 34 - p. 4, line 3):

"The LCST of the P(NIPAAm-co-AAcNa) hydrogels in water increased with increasing AAcNa content and disappeared by introducing an AAcNa content of more than 5 mol% (Figure 2a). This means that the introduction of AAcNa component to enhance the moisture absorption capacity is limited to 5 mol%. In contrast, a clear LCST for the PNIPAAm/Alg IPN hydrogel was observed at 32 °C, which is the same as the LCST of PNIPAAm, despite it containing a hydrophilic Alg component. The introduction of the IPN structure enhanced the hydrophilicity of the networks without changing the LCST of PNIPAAm. As a result, the PNIPAAm/Alg IPN hydrogel exhibited a drastically thermo-responsive change in hydrophilicity/hydrophobicity with enhancing the moisture absorption capacity, differing from PNIPAAm-containing copolymer hydrogels."

Thank you again for your comments on our paper. We considered your kind advices and revised our manuscript. I hope this revision will meet your requirements satisfactory.

<Corresponding explanation for the reviewer's comments (Reviewer #3)>

Thank you very much for your kind comments on our manuscript. We revised it under your comments. We would like to explain which changes according to the reviewer's comments were made as follows:

Reviewer's comment (1):

In this work, the authors developed novel water gas condensation system by interpenetrating polymer network consisting of thermo-responsive polymer. Although the NIPAAm-based hydrogels were well investigated, this system has never been developed. This system is a unique and new type of gas condensation system, and has potential impact to broad range of readers. However, one important point remains to be untackled in the current version; that is, repetitive absorption-oozing cycle. When the authors confirm the ability of repetitive absorption-oozing cycle, I recommend it for publication in Nature Communications.

Response:

As pointed out by the reviewer #3, repetitive absorption-oozing cycle is the most important for developments of innovative high-efficiency energy exchange systems on the temperature-controlled water absorption/oozing concept. Therefore, we performed repetitive cycle tests between moisture adsorption and water oozing using the dried PNIPAAm/Alg IPN gel. The 15-cycle tests revealed that the dried PNIPAAm/Alg IPN gels absorbed water from the gas phase below its LCST, and that then, upon heating to slightly above the LCST, the gel oozed liquid water repeatedly. Together with Figure 4d, we added the sentences about the repetitive cycle tests in the main text and Methods.

In the main text (p. 6, lines 3-7):

"In addition, we performed cycle tests of moisture absorption and water oozing using the IPN gel because its applications to energy exchange systems such as dehumidifiers with low energy consumption require repetitive water oozing. The cycle tests demonstrated that moisture absorbed into the IPN gel repeatedly oozed out as liquid water during cyclic changes between the temperature below and above LCST (Figure 4d)."

In Methods (p. 8, lines 24-31):

"Cycle tests of moisture absorption and water oozing

Cycle tests for moisture absorption and water oozing were performed using the dried PNIPAAm/Alg IPN gel with a diameter of 70 mm and a thickness of 1.5 mm as follows. First of all, moisture was absorbed into the dried PNIPAAm/Alg IPN gel at 27 °C and 80% RH. After the moisture absorption for 60 min., the IPN gel was placed on a plate heater at 50 °C. Then the water oozed from the surface of the IPN gel was transferred to the surface of a glass plate and was its amount was determined by measuring a change in the weight of the glass plate, completing one full test cycle. This procedure was repeated 15 times with the same IPN gel."

Reviewer's comment (2):

Another minor point is P(NIPAAm-co-AAc) gels in Figure 2a. Because there are few description on these samples, I recommend remove them or add more experiments and discussion on these samples.

Response:

As pointed out by the reviewer #3, there are few descriptions on the P(NIPAAm-co-AAc) gels in the manuscript. According to the reviewer's recommendation, we added the sentences about experiments and discussion on the P(NIPAAm-co-AAc) hydrogels to clarify the importance of IPN structure by comparing PNIPAAm/Alg IPN and P(NIPAAm-co-AAc) hydrogels as follows (p. 3, line 34 - p. 4, line 3):

"The LCST of the P(NIPAAm-co-AAcNa) hydrogels in water increased with increasing AAcNa content and disappeared by introducing an AAcNa content of more than 5 mol% (Figure 2a). This means that the introduction of AAcNa component to enhance the moisture absorption capacity is limited to 5 mol%. In contrast, a clear LCST for the PNIPAAm/Alg IPN hydrogel was observed at 32 °C, which is the same as the LCST of PNIPAAm, despite it containing a hydrophilic Alg component. The introduction of the IPN structure enhanced the hydrophilicity of the networks without changing the LCST of PNIPAAm. As a result, the PNIPAAm/Alg IPN hydrogel exhibited a drastically thermo-responsive change in hydrophilicity/hydrophobicity with enhancing the moisture absorption capacity, differing from PNIPAAm-containing copolymer hydrogels."

Thank you again for your comments on our paper. We considered your kind advices and revised our manuscript. I hope this revision will meet your requirements satisfactory.

Reviewers' comments:

Reviewer #1 (Remarks to the Author):

Dear authors: although it is true that most of the research on hydrogels is focused in the changes from the wet state to the dried state, I still consider that the findings in this manuscript does not exhibit enough novelty to be published in Nature Communications. The fact that the research is focused in the dried state of the hydrogels, only changes the starting point of the experiment in a possible cycling experiment. In my opinion this is a very interesting work but lacks the novelty for Nature Communications.

Reviewer #2 (Remarks to the Author):

The authors addressed satisfactorily some of the points raised by the reviewer. However, several critical concerns seem to remain and are still not clearly explained.

1. Concerning the liquid condensation mechanism in the IPN gel, the authors replied with liquid-vapor equilibrium. However, the authors' answer is insufficient to explain the liquid-vapor phase transition of water in the IPN gel. The authors need to explain the origin of phase transition of water with change in molecular conformation of the PNIPAAm and Alg in the IPN gel.

2. The authors replied that the amount of oozed water is about 10% of total absorbed vapor. It is necessary to add such information in the revised manuscript. And the reviewer recommends to describe the oozed water/vapor ratio in the experimental conditions. In Figure 2c, although the content of water absorption exhibited the significant difference in response to the temperature variation, only 10% of absorbed water was oozed from the IPN gel, which means that 90% of absorbed water remained in the IPN gel. In the revised manuscript, the authors did not mention remained water inside the IPN gel. The authors need to explain how that large amount of remaining water is preserved in the IPN gel after oozing water.

3. The authors discussed about absorption model depending on the hydrophilic/phobic behavior of the IPN gel. However, this discussion is insufficient to explain the hydrophilic/phobic blend system. The contact angles of probe liquids such as water are required to explain the change in surface energy of the blend gel in response to the temperature change. The references the authors provided described the swelling of hydrophilic polymers. The authors need to provide the applicability of the above description to hydrophilic/phobic blend systems, and explain the mechanism of transition in hydrophilicity of IPN gel with temperature in molecular level.

Reviewer #3 (Remarks to the Author):

Because the authors have well responded to the comments, I recommend accepting this manuscript for publication.

<Corresponding explanation for the reviewer's comments (Reviewer #1)>

Thank you very much for your kind comments on our manuscript. We revised it under your comments. We would like to explain which changes according to the reviewer's comments were made as follows:

Reviewer's comment (1):

Dear authors: although it is true that most of the research on hydrogels is focused in the changes from the wet state to the dried state, I still consider that the findings in this manuscript does not exhibit enough novelty to be published in Nature Communications. The fact that the research is focused in the dried state of the hydrogels, only changes the starting point of the experiment in a possible cycling experiment. In my opinion this is a very interesting work but lacks the novelty for Nature Communications.

Response:

As commented by the reviewer #1, our concept includes a simple change of the starting point in a possible cycling experiment. Standard hydrophilic polymers can absorb moisture from air but can't ooze it as liquid water. To generate the hydrophilic polymers after moisture absorption, they must be dried by evaporation of absorbed water under high temperature. Condensation of evaporated water by cooling is also required to collect moisture as liquid water. On the other hand, although our concept includes the simple change from a wet state to a dried state in the starting point, it enabled us to directly collect liquid water from moisture by small changes in temperature. By the simple change of the starting point, we have achieved the collection of moisture (gaseous water) as liquid water using thermo-responsive gel. Nobody has succeeded in the collection of moisture by a small change in temperature. In addition, the studies on thermo-responsive gels have utilized in wet states and their responsive behaviors have been observed in aqueous solutions. This is the first paper describing innovative applications of thermo-responsive gels in a dried state. In this study, we succeeded in quantifying the unique behavior of thermo-responsive gels in a dried state by measuring moisture absorption kinetics and amount of water oozed from the gels. Therefore, we believe that the simple change of the starting point of the experiment is an important finding, demonstrating several potential applications of thermo-responsive gels to humidifiers and so on. As the reviewer's comment about the starting point is very important, we added the description about our concept that we can achieve the collection of moisture as a liquid water using thermo-responsive gel by a simple change of the starting point in a possible cycling experiment in the revised manuscript as follows (p. 3, lines 26-30):

"Our concept includes a simple change of starting point from a wet state to a dried state in applications of thermo-responsive gels. By the simple change of the starting point, we can achieve the collection of moisture (gaseous water) as liquid water using thermo-responsive gel, followed by innovative applications of thermo-responsive gels in their dried state."

Thank you again for your comments on our paper. We considered your kind advices and revised our manuscript. I hope this revision will meet your requirements satisfactory.

<Corresponding explanation for the reviewer's comments (Reviewer #2)>

Thank you very much for your kind comments on our manuscript. We revised it under your comments. We would like to explain which changes according to the reviewer's comments were made as follows:

Reviewer's comment (1):

The authors addressed satisfactorily some of the points raised by the reviewer. However, several critical concerns seem to remain and are still not clearly explained.

Concerning the liquid condensation mechanism in the IPN gel, the authors replied with liquid-vapor equilibrium. However, the authors' answer is insufficient to explain the liquid-vapor phase transition of water in the IPN gel. The authors need to explain the origin of phase transition of water with change in molecular conformation of the PNIPAAm and Alg in the IPN gel.

Response:

It is difficult to explain the detailed mechanism of the liquid-vapor phase transition of water in the IPN gel. However, in the revised manuscript, we try to explain the mechanism from moisture absorption onto hydrophilic chains, a drastic change in the hydrophilic/hydrophobic of PNIPAAm chains and a slight swelling/shrinkage of the networks as follows. During the moisture absorption into the IPN gel below the LCST, water molecules are adsorbed on both Alg and PNIPAAm chains similarly to the moisture absorption into general hydrophilic polymers, followed by a slight swelling of the IPN gel. The slight swelling results in hydrophobic hydration of PNIPAAm chains although the gel network is not immersed in an aqueous solution. It is well known that PNIPAAm undergoes a drastic change from hydrophilic to hydrophobic in aqueous solutions with rising temperature above the LCST because the negative entropy of water molecules around the nonpolar regions of PNIPAAm chains dominates. Similarly, the PNIPAAm chains in the IPN gel slightly swollen by moisture absorption change from hydrophilic to hydrophobic by the dehydration based on the negative entropy of water molecules around the PNIPAAm chains with rising temperature. As the PNIPAAm chains become hydrophobic above the LCST, the water molecules on the PNIPAAm chains are desorbed to be condensed as liquid water. Thus, the moisture absorbed into the thermo-responsive IPN gel oozes out as liquid water above the LCST. In contrast, the water molecules adsorbed on the Alg chains are preserved within the IPN gel owing to their high hydrophilicity despite a temperature above the LCST of PNIPAAm. Desorption of water molecules from the PNIPAAm chains results in a slight shrinkage of the gel network. Water molecules on the Alg chains may be squeezed out by the slight shrinkage of the network with oozing water from the PNIPAAm chains. As a result, the thermo-responsive IPN gel can absorb moisture from air and oozing it as liquid water by cycle changes in temperature.

Thus, according to the reviewer's comments, we added the description about the possible mechanism as follows (p. 6, lines 33-P. 7, line 13):

"From the discussions about moisture absorption using Langmuir and BET type models and about the activation energy for water release above the LCST, we can explain the mechanism for the absorption of moisture into the dried IPN gel and oozing it as liquid water as follows. During the moisture absorption into the IPN gel below the LCST, water molecules are adsorbed on both Alg and PNIPAAm chains similarly to the moisture absorption into general hydrophilic polymers¹⁶⁻¹⁸, followed by a slight swelling of the IPN gel. The slight swelling results in hydrophobic hydration of PNIPAAm chains although the gel network is not immersed in an aqueous solution. It is well known that PNIPAAm undergoes a drastic change from hydrophilic to hydrophobic in aqueous solutions with rising temperature above the LCST because the

negative entropy of water molecules around the nonpolar regions of PNIPAAm chains dominates. Similarly, the PNIPAAm chains in the IPN gel slightly swollen by moisture absorption change from hydrophilic to hydrophobic by the dehydration based on the negative entropy of water molecules around the PNIPAAm chains with rising temperature. As the PNIPAAm chains become hydrophobic above the LCST, the water molecules on the PNIPAAm chains are desorbed to be condensed as liquid water. Thus, the moisture absorbed into the thermo-responsive IPN gel oozes out as liquid water above the LCST. In contrast, the water molecules adsorbed on the Alg chains are preserved within the IPN gel owing to their high hydrophilicity despite a temperature above the LCST of PNIPAAm. Desorption of water molecules from the PNIPAAm chains results in a slight shrinkage of the gel network. Water molecules on the Alg chains may be squeezed out by the slight shrinkage of the network with oozing water from the PNIPAAm chains. As a result, the thermo-responsive IPN gel can absorb moisture from air and oozing it as liquid water by cycle changes in temperature."

Reviewer's comment (2):

The authors replied that the amount of oozed water is about 10% of total absorbed vapor. It is necessary to add such information in the revised manuscript. And the reviewer recommends to describe the oozed water/vapor ratio in the experimental conditions. In Figure 2c, although the content of water absorption exhibited the significant difference in response to the temperature variation, only 10% of absorbed water was oozed from the IPN gel, which means that 90% of absorbed water remained in the IPN gel. In the revised manuscript, the authors did not mention remained water inside the IPN gel. The authors need to explain how that large amount of remaining water is preserved in the IPN gel after oozing water.

Response:

In the previously revised manuscript, we described that the amount of oozed water was approximately 10wt% of total vapor water absorbed into the thermos-responsive IPN gel. Again we carried out cycle tests carefully and collected approximately 20% water as a liquid state to the totally absorbed moisture (weight of collected water/weight of absorbed moisture = ca. 0.2) from the gel, as shown in Figure 4d. However, this mean that 80% of absorbed water still remained in the IPN gel. At a temperature above the LCST, as PNIPAAm chains change from hydrophilic to hydrophobic but Alg chains are hydrophilic, hydrophilic Alg chains prevented water from being released out of the IPN gel. Therefore, not all water molecules can ooze out as a liquid water from the IPN gel because of their adsorption on the hydrophilic Alg chains even if PNIPAAm chains become hydrophobic above the LCST. In addition, water absorbed into the IPN gel is partially released as a vapor state, based on vapor–liquid equilibrium under the condition. As a result, we cannot collect all water molecules absorbed in the IPN gel but can collect approximately 20wt% water as a liquid state. This is the reason why absorbed water partially remains in the IPN gel after water oozing out above the LCST.

The reviewer gave us a good suggestion that we should add the information about the amount of oozed water to totally absorbed moisture. According to the reviewer's suggestion, the figure about the weight of ooze water was exchanged with that about the weight ratio of oozed water to totally absorbed moisture in Figure 4d. In addition, according to the reviewer's comment, we added the discussion about the remaining water inside the IPN gel as follows (p. 6, lines 22-32):

"In the cycle tests using the IPN gel, we collected approximately 20% water as a liquid state to the totally absorbed moisture, meaning that 80% water still remained in the gel. At a temperature above the LCST, as PNIPAAm chains change from hydrophilic to hydrophobic but Alg chains are hydrophilic, hydrophilic

Alg chains prevented water from being released out of the IPN gel. Therefore, not all water molecules can ooze out as a liquid water from the IPN gel because of their adsorption on the hydrophilic Alg chains even if PNIPAAm chains become hydrophobic above the LCST. In addition, water absorbed into the IPN gel is partially released as a vapor state, based on vapor–liquid equilibrium under the condition. As a result, we cannot collect all water molecules absorbed in the IPN gel but can collect approximately 20wt% water as a liquid state. This is the reason why absorbed water partially remains in the IPN gel after water oozing out above the LCST. "

Reviewer's comment (3):

The authors discussed about absorption model depending on the hydrophilic/phobic behavior of the IPN gel. However, this discussion is insufficient to explain the hydrophilic/phobic blend system. The contact angles of probe liquids such as water are required to explain the change in surface energy of the blend gel in response to the temperature change. The references the authors provided described the swelling of hydrophilic polymers. The authors need to provide the applicability of the above description to hydrophilic/phobic blend systems, and explain the mechanism of transition in hydrophilicity of IPN gel with temperature in molecular level.

Response:

It is well known that PNIPAAm chains undergo a drastic change from hydrophilic to hydrophobic with rising temperature above the LCST. However, as pointed out by the reviewer, we should demonstrate a change in the hydrophilicity of the PNIPAAm/Alg IPN gel by contact angle measurements. Therefore, we measured water contact angles on the dried IPN gel below and above its LCST. The IPN gel showed small water contact angles at 25 and 50 °C owing to hydrophilic Alg chains. The water contact angle on the IPN gel increased from 23.1° to 36.3 ° by rising temperature from 25 to 50 °C above the LCST, as shown in Figure S1. The larger water contact angle at 50 °C than 25 °C demonstrates that the hydrophobicity of the dried IPN gel increased by rising temperature in air. Although hydrophilic Alg chains in the IPN gel prevent the water contact angle from increasing remarkably, a drastic change of PNIPAAm chains from hydrophilic to hydrophobic above the LCST causes an increase in water contact angle on the IPN gel in air. Therefore, water oozing from the IPN gel is mainly attributed to a drastic change in the hydrophilicity/hydrophobicity of the thermo-responsive PNIPAAm chains. We added Figure S1 about the water contact angles on the IPN gel in Supporting Information and the discussion about hydrophilicity change in the text as follows (p. 5, lines 24-34):

"To reveal a change in the hydrophilicity of the dried IPN gel, we measured water contact angles on the gel at 25 and 50 °C. The IPN gel showed small water contact angles at 25 and 50 °C owing to hydrophilic Alg chains. The water contact angle on the IPN gel increased from 23.1° to 36.3 ° by rising temperature from 25 to 50 °C above the LCST (Figure S1). The larger water contact angle at 50 °C than 25 °C demonstrates that the hydrophobicity of the dried IPN gel increased by rising temperature in air. Although hydrophilic Alg chains in the IPN gel prevent the water contact angle from increasing remarkably, a drastic change of PNIPAAm chains from hydrophilic to hydrophobic above the LCST causes an increase in water contact angle on the IPN gel in air. Therefore, water oozing from the IPN gel is mainly attributed to a drastic change in the hydrophilicity/hydrophobicity of the thermo-responsive PNIPAAm chains."

In addition, according to the reviewer's suggestion, we added the explanation about the mechanism for a change in hydrophilicity of the IPN Gel with temperature in molecular level as follows (p. 6, line 35 - p. 7, line 6):

"During the moisture absorption into the IPN gel below the LCST, water molecules are adsorbed on both Alg and PNIPAAm chains similarly to the moisture absorption into general hydrophilic polymers¹⁶⁻¹⁸, followed by a slight swelling of the IPN gel. The slight swelling results in hydrophobic hydration of PNIPAAm chains although the gel network is not immersed in an aqueous solution. It is well known that PNIPAAm undergoes a drastic change from hydrophilic to hydrophobic in aqueous solutions with rising temperature above the LCST because the negative entropy of water molecules around the nonpolar regions of PNIPAAm chains dominates. Similarly, the PNIPAAm chains in the IPN gel slightly swollen by moisture absorption change from hydrophilic to hydrophobic by the dehydration based on the negative entropy of water molecules around the PNIPAAm chains with rising temperature. As the PNIPAAm chains become hydrophobic above the LCST, the water molecules on the PNIPAAm chains are desorbed to be condensed as liquid water."

Thank you again for your comments on our paper. We considered your kind advices and revised our manuscript. I hope this revision will meet your requirements satisfactory.

<Corresponding explanation for the reviewer's comments (Reviewer #3)>

Thank you very much for your kind comments on our manuscript. We are very pleased to hear that the reviewer #3 recommended accepting our manuscript for publication in Nature Communications. The reviewer's comments have improved our manuscript. Again we would like to appreciate for your kind advises.

REVIEWERS' COMMENTS:

Editorial Note: Reviewer #3 was asked to look over the authors' response to Reviewer #2's comments from the last round. Reviewer #3 made comments to the editor and suggested that it may be better to focus less on the amount of water oozed-out from hydrogels, and instead highlight the strong point of this paper, i.e., "collecting and releasing water with small temperature change".

Reviewer #3 (Remarks to the Author):

The authors well responded to the comment of Reviewer #2.
I recommend publishing this paper as is.

<Corresponding explanation for the reviewer's comments (Reviewer #3)>

Thank you very much for your kind comments on our manuscript. We revised it under your comments. We would like to explain which changes according to the reviewer's comments were made as follows:

Reviewer's and editor's comment (1):

Reviewer #3 was asked to look over the authors' response to Reviewer #2's comments from the last round. Reviewer #3 made comments to the editor and suggested that it may be better to focus less on the amount of water oozed-out from hydrogels, and instead highlight the strong point of this paper, i.e., "collecting and releasing water with small temperature change".

Response:

We agree to the reviewer's comment to the editor. Therefore, although the previous manuscript included lots of discussion about the amount of water oozed-out from hydrogels, the discussion is simplified in the revised manuscript. In addition, to focus on unique thermo-responsive behaviour, "collecting and releasing water with small temperature change", we revised the manuscript by modifying sentences and adding the phrase, "by a small temperature change" and an equivalent phrase. Thus, according to the reviewer's comments, we revised our manuscript to focus less on the amount of water oozed-out from hydrogels and to highlight the strong point of this paper, i.e., "collecting and releasing water with small temperature change".

Reviewer's comment (2):

The authors well responded to the comment of Reviewer #2. I recommend publishing this paper as is.

Response:

Thank you very much for your kind comments on our manuscript. We are very pleased to hear that the reviewer #3 recommended accepting our manuscript for publication in Nature Communications. The reviewer's comments have improved our manuscript. Again, we would like to appreciate for your kind advises.

Thank you again for your comments on our paper. We considered your kind advices and revised our manuscript. I hope this revision will meet your requirements satisfactory.